# Comparison of Temporomandibular Disorder Signs and Symptoms in CrossFit^®^ Athletes and Sedentary Individuals

**DOI:** 10.3390/ijerph22050785

**Published:** 2025-05-16

**Authors:** Ana Paula Varela Brown Martins, Ranele Luiza Ferreira Cardoso, Caio César Ferreira Versiani de Andrade, Júlia Meller Dias de Oliveira, Maria Beatriz Freitas D’Arce, Adriana Barbosa Ribeiro, Carolina Noronha Ferraz de Arruda, Juliana Silva Ribeiro de Andrade, Bianca Miarka, Maurício Malheiros Badaró

**Affiliations:** 1Department of Dentistry, Federal University of Santa Catarina, Florianópolis Campus, Florianópolis 88040-535, SC, Brazil; apvbmartins@gmail.com (A.P.V.B.M.); julia_meller5@hotmail.com (J.M.D.d.O.); sribeirooj@gmail.com (J.S.R.d.A.); 2Department of Dentistry, Federal University of Juiz de Fora, Governador Valadares Campus, Governador Valadares 35020-220, MG, Brazil; ranelecrds@gmail.com (R.L.F.C.); caio.cesarandrade@hotmail.com (C.C.F.V.d.A.); 3Department of Prosthodontics, Federal University of Espírito Santo, Vitória 29040-090, ES, Brazil; maria.darce@ufes.br; 4Department of Dental Materials and Prosthesis, Ribeirao Preto School of Dentistry, University of Sao Paulo, Ribeirao Preto 14040-904, SP, Brazil; driribeiro@usp.br; 5Department of Prosthodontics, Dental School, Rio de Janeiro State University, Rio de Janeiro 20551-030, RJ, Brazil; carolina.arruda@uerj.br; 6Department of Combat Sports, School of Physical Education and Sports, Federal University of Rio de Janeiro, Rio de Janeiro 21941-599, RJ, Brazil; miarkasport@hotmail.com

**Keywords:** temporomandibular joint disorders, myofascial pain syndromes, temporomandibular joint, exercise, sports

## Abstract

(1) Background: A sedentary lifestyle may aggravate temporomandibular disorder (TMD) symptoms, increasing pain sensitivity and functional limitations. Physical exercise is recommended for pain management and improving quality of life. Comparing CrossFit^®^ athletes to sedentary individuals allows for examining whether regular high-intensity exercise impacts pain sensitivity and functional limitations associated with TMD. This cross-sectional study assessed the signs and symptoms of TMD in CrossFit^®^ athletes compared to sedentary individuals. (2) Methods: Participants (n = 121) were divided into four groups: sedentary with TMD (n = 39), sedentary without TMD (n = 37), CrossFit^®^ athletes with TMD (n = 23), and CrossFit^®^ athletes without TMD (n = 22). TMD signs and symptoms were evaluated using the Research Diagnostic Criteria for TMD (RDC/TMD) axis I, including mandibular movement patterns, range of motion, joint sounds, muscle pain, and jaw dysfunctions. Statistical analyses included chi-square and Dunn’s post hoc tests, ANOVA, and Kruskal–Wallis tests. Correlation and regression analyses were performed to examine associations between CrossFit^®^ practice and TMD (*p* ≤ 0.05). (3) Results: Myofascial pain was the most common diagnosis. All athlete groups exhibited greater mandibular movement amplitudes (unassisted opening without pain, *p* < 0.001, and protrusion, *p* = 0.039) and less pain (*p* < 0.001) than sedentary individuals. Pain reports and palpation-induced pain in muscles and joints were significantly associated with a sedentary lifestyle and TMD (*p* < 0.001). Joint and muscle pain were more prevalent (frequent) among sedentary participants, regardless of TMD diagnosis. Linear regression analysis showed that sedentary individuals without TMD had significantly reduced unassisted mouth opening amplitudes compared to athletes without TMD (*p* < 0.05). (4) Conclusions: Pain in the masseter, temporalis, posterior digastric, and medial pterygoid muscles was the most common symptom in sedentary individuals with TMD. They experience higher frequency and intensity of pain, as well as greater limitations in mouth movement. Athletes showed higher frequency of joint noises.

## 1. Introduction

Temporomandibular disorders (TMDs) are a heterogeneous group of conditions, that is, it includes several clinical conditions (pain localized in muscles of mastication or preauricular area), each requiring distinct management approaches, which is aggravated with chewing or other mandibular activities or movements [1,2,3]. These disorders can significantly impact individuals’ quality of life and have a multifactorial etiology, manifesting signs and symptoms due to both pathophysiological (systemic, local, and genetic factors) and psychosocial (stress, anxiety, limited coping skills, depression, etc.) factors [2,4]. The prevalence of TMD is approximately 31% among adults and older adults, and 11% in children and adolescents [3]. The most common signs and symptoms of TMD include facial pain, muscle and/or temporomandibular joint (TMJ) pain, limited or altered mandibular movements, and joint noises [2,5]. In some cases, these can lead to severe limitations and disability [6,7,8] and decreasing labor productivity [9]. Additionally, TMD can also be associated with unusual symptoms such as earache, tinnitus, chewing difficulties, and dizziness [10]. While the prevalence of TMD in the general population has been explored [3,11], there exists a research gap regarding its frequency and impact among athletes, especially those engaged in high-intensity sports like CrossFit^®^.

Cross-discipline fitness, or CrossFit^®^ is a training program comprising intermittent, cyclical tasks of varying intensities, including weightlifting, plyometric activities, calisthenics, running, and indoor rowing [12,13]. Practicing CrossFit^®^ requires an advanced-level technique due to maximum repetitions of timed exercises without adequate rest intervals between sets, and insufficient recovery time between high-volume loads and training sessions [14]. The overload resulting from this exercise pattern can lead to early fatigue, additional oxidative stress, less resistance to the subsequent repetitive strain of exercise, greater perception of effort, and unsafe execution of movements [14]. Among these injuries, there is a disproportionate risk of musculoskeletal injuries, especially for novice participants, resulting in loss of service time, medical treatment, and extensive rehabilitation [14].

Athletes often face intense daily physical, competitive sports, and psychological pressures [15,16]. Athletes are more likely to develop and sustain chronic pain conditions due to high-intensity training volumes, repetitive movements, orthopedic surgery, stress fracture, and muscular imbalance [17,18,19]. The incidence density of CrossFit^®^-related musculoskeletal injuries was 18.9 per 1000 h of exposure [13]. Most oral behaviors in the waking state often occur unconsciously, making them difficult to identify [20]. Possibly, CrossFit^®^ athletes, while performing repetitive movements that lead to exhaustion, may acquire harmful oral behaviors that can be considered etiological factors for TMD (bruxism, jaw muscle fatigue, improper posture, and increased stress). Prior research has indicated that chronic pain conditions, such as TMD, can significantly affect athletes’ performance, potentially exacerbated by high training volumes and repetitive movements [18,21].

However, detailed studies focusing on the frequency and impact of signs and symptoms of TMD in sports practitioners are scarce [5,19,21], especially in emerging fitness trends like CrossFit^®^. Understanding the frequency of TMD considering clinical aspects can help in developing targeted preventive measures and individualized treatments, ultimately improving athletes’ performance and well-being. Multi-professional support for athletes is vital to implement preventive interventions against potential injuries. Therefore, the aim was to assess the frequency of signs and symptoms of TMD in CrossFit^®^ practitioners and, to compare these findings with those from sedentary individuals. This comparison will offer insights into how high-intensity physical activity influences the TMD. We hypothesize that the frequency and severity of TMD signs and symptoms will be different between these two groups, potentially offering new perspectives on how physical activity impacts TMD.

## 2. Materials and Methods

### 2.1. Study Design

This cross-sectional study was descriptive with four groups, comparing high-performance athletes and sedentary individuals with and without TMD. The ethics committee at the Clementino Fraga Filho University Hospital of the Federal University of Rio de Janeiro approved this study (CAAE: 13846919.8.0000.5257, 25 November 2019). A non-probabilistic convenience sampling method was employed for participant recruitment. Initially, the participants were recruited verbally, and once they had accepted, they were contacted virtually via e-mail for physical assessment. All volunteers were informed about this study’s nature and clinical procedures and signed a consent form. They gave their informed consent following the Helsinki Declaration, understanding that they were free to withdraw from the study at any time.

The sample was composed of athletes. Data collection for high-performance athletes was conducted in four gyms accredited by CrossFit^®^: Brutus CrossFit^®^, Caverna CrossFit^®^, CrossFit^®^ Órbita, and CrossFit^®^ Z1, all located in Governador Valadares/MG, Brazil (map.crossfit.com accessed on April 8, 2019). The data collection for non-CrossFit^®^ practitioners (sedentary individuals) was carried out among individuals seeking dental care at the Federal University of Juiz de Fora, Governador Valadares campus (UFJF-GV), also in Governador Valadares/MG. The period of recruitment ranged from 2019 to 2020.

The inclusion criteria for all participants were age between 18 and 48 years; absence of rheumatological conditions, neurological disorders, motor impairments, primary headaches, or orofacial inflammatory/infectious conditions [20]; no regular use of medication; no toothache, polyarthritis, or other rheumatic diseases; and provision of informed consent. For participants with temporomandibular disorders (TMDs), only those diagnosed with muscular TMD—with or without associated joint TMD—were included. Sedentary individuals were defined as those who had not engaged in any amateur or professional sport or physical activity for at least six months. Sedentary behavior was characterized as low energy expenditure while awake, in various settings such as work, school, home, community, or during transportation [22]. CrossFit^®^ practitioners were required to have competed in regional and/or national competitions, trained at least four times per week for a minimum of six months, and attended an accredited facility under the supervision of a certified coach. Their training regimen involved intermittent and cyclical exercises of varying intensities, including weightlifting, plyometrics, calisthenics, running, and indoor rowing [12].

The exclusion criteria included isolated joint TMD, recent mandibular fracture, current dental or physical therapy treatment that could affect TMD, neurological impairment, previous treatment for TMD, and use of painkillers and/or anti-inflammatory drugs within 48 h before data collection. After applying the eligibility criteria, groups were formed with sedentary individuals without and with TMD, and athletes in the same condition.

### 2.2. Measurements and Procedures

To minimize bias in this clinical study, observers were standardized in their evaluation procedures and data collection methods. They were blinded to the data distribution during tabulation, ensuring impartiality in the analysis. Additionally, a researcher from a non-dental field was selected to further reduce potential bias, as they had no prior knowledge of the meaning of the collected data, the study outcomes, or the hypothesis under investigation. Furthermore, the results were analyzed separately for different subgroups within the total sample, enhancing the reliability and robustness of the findings.

The condition of blinding was due to the differentiation in roles between the researchers. Three examiners were responsible for different stages, with no contact between them. A first examiner (M.M.B.) screened the participants in terms of whether they practiced sports or were sedentary. A second researcher (R.L.F.C.) applied the diagnostic system RDC/TMD axis I (clinical examination), while another researcher (A.P.V.B.M.) carried out the diagnosis. While processing the data collected, athletes and sedentary people were assigned random numbers for statistical analysis. Blinding of the second examiner was not completely feasible due to data collection occurring at the CrossFit^®^ athletes’ training facilities and the evident physical disparities between the athlete and sedentary groups. However, the examiner was unaware of the TMD diagnosis among both CrossFit^®^ and sedentary individuals. Prior to this study, the examiner underwent calibration using a standardized scale to ensure consistency in muscle palpation during the physical examination.

Data collection was conducted in the morning for all volunteers. Materials such as disposable gauze, tongue depressors, gloves, a stethoscope, a millimeter ruler, and a traditional bow compass were utilized during the clinical examination. Volunteers experiencing more severe pain were referred to a specialized dental clinic at UFJF-GV, where they received conservative treatments based on their diagnosis.

The Research Diagnostic Criteria for Temporomandibular Disorders (RDC/TMD) is used for clinical assessment, diagnosis, and categorization of TMD. It is based on a biobehavioral model of pain, consisting of two main axes: signs and symptoms (axis I) and psychological and disability factors (axis II) [9]. Anamnestic data collection and clinical examination were conducted following the RDC/TMD guidelines and carried out by a single operator (R.L.F.C.), who was previously calibrated by experts (A.P.V.B.M. and M.M.B.). The RDC/TMD diagnostic system was applied in its validated Portuguese version [23]. The RDC/TMD examinations enabled the diagnosis of muscle disorders (group I), disc displacement (group II), and arthralgia, osteoarthritis, and osteoarthrosis (group III) [24]. The RDC/TMD indicates that the same individual can be diagnosed with more than one subtype of TMD [25].

Axis I of the RDC/TMD requires a physical exam and algorithmic diagnosis of muscle pain; according to Bonotto et al. [25]: The diagnosis of myofascial pain is confirmed when more than three ipsilateral muscle sites are tender on palpation. The RDC/TMD algorithms for pain are used to diagnose joint pain associated with TMD. In cases of disc displacement with arthralgia, pain is considered present when the joint palpation and clicking result in pain on the same side of the joint (without elimination by protruding opening), or on the same side of the deviation with reduced opening, where the difference between unassisted opening and assisted opening is less than 5 mm. The presence of joint pain associated with coarse crepitus indicates the presence of osteoarthritis. Symptoms frequently associated with TMD, such as any joint noises (clicking), crepitus detected (crackling), teeth clenching, uncomfortable bite, stiffness, weakness, fatigue muscle, and ear symptoms (tinnitus), were also observed. In this study, axis I of RDC/TMD was used.

Before starting this research, all examiners were trained and calibrated according to the standards present on the official website of the RDC/TMD International Consortium [26].

### 2.3. Statistical Analysis

Continuous quantitative variables in this study were tested for normality and found to not have a normal distribution, as indicated by the Shapiro–Wilk test (*p* < 0.05). Descriptive statistics were calculated for all demographic and clinical variables, including means, standard deviations, and percentages. Between-group comparisons (sedentary vs. CrossFit^®^ athletes) were performed using chi-square tests (χ^2^) for categorical variables and independent t-tests or Mann–Whitney U tests for continuous variables, depending on data normality. Normality was assessed using the Shapiro–Wilk test.

To evaluate differences in mandibular function, pain reporting, and joint noises across groups (athletes vs. sedentary, with and without TMD), one-way ANOVA was applied for normally distributed continuous variables, while Kruskal–Wallis tests were used for non-parametric comparisons. Post hoc pairwise comparisons were performed using Bonferroni correction to adjust for multiple comparisons [27] and control the Type I error rate.

A multiple regression analysis and a binary logistic regression analysis were conducted to assess the association between physical activity status (sedentary vs. athlete) and the presence of TMD, as well as the association between physical activity status and self-reported pain (binary outcome: presence vs. absence of pain), respectively. The model included age and gender as covariates, and the strength of association was expressed as odds ratios (ORs) with 95% confidence intervals (CIs). The pseudo R^2^ value was reported to indicate the proportion of variance explained by the model. The Likelihood Ratio Test (LLR test, *p* < 0.001) confirmed the model’s statistical significance.

All statistical tests were two-tailed, and effect sizes were calculated where applicable to provide additional insights into the clinical relevance of findings. All statistical analyses were performed using SPSS Statistics for Windows, version 26.0 (SPSS Inc., Chicago, IL, USA). A significance level of *p* < 0.05 was adopted for all tests.

## 3. Results

The final sample comprised 121 participants (Figure 1), of whom 58.67% were female (athletes without TMD: 45; sedentary without TMD: 55.88; athletes with TMD: 48; and sedentary with TMD: 77.77). The average age was ± 29 years. Most sedentary individuals were aged 18 to 27 (without TMD: 68.75% and with TMD: 72.23%). For athletes, both groups showed 40% in the 28 to 37 age range. The majority of participants were single (72.25%) and had incomplete higher education. Most sedentary individuals had incomplete higher education (without TMD: 64.7% and with TMD: 78.57%). Among athletes, 70% of those without TMD had completed higher education, while 60% of those with TMD had completed high school. The study population was divided into sedentary individuals (n = 76) and CrossFit^®^ athletes (n = 45).

Among sedentary participants, 51.3% were diagnosed with temporomandibular disorder (TMD), whereas 51.1% of CrossFit^®^ athletes presented with TMD, indicating a comparable prevalence between groups. However, sedentary individuals were 2.3 times more likely to develop TMD than athletes after controlling for age and gender (χ^2^ = 10.24, OR = 2.3, *p* = 0.012).

The presence of deviation or deflection was high across all groups, with no significant differences between sedentary and athletic individuals (*p* = 0.13) (Table 1). However, joint noises differed significantly: athletes with TMD exhibited a higher frequency of opening noises (*p* < 0.001), while lateral motion noises (*p* < 0.001) and protrusion noises (*p* = 0.002) were significantly more frequent in sedentary individuals with TMD. Closing noises did not significantly differ across groups after multiple comparison correction (*p* = 0.069).

Analysis of mandibular movement revealed that unassisted opening without pain was significantly greater in athletes without TMD compared to sedentary individuals (*p* < 0.001) (Table 2). Similarly, protrusion values were lower in sedentary individuals with TMD compared to athletes without TMD (*p* = 0.039). However, maximum unassisted and assisted opening did not significantly differ between groups (*p* > 0.05).

Self-reported pain and muscle palpation responses were significantly higher in sedentary individuals with TMD (*p* < 0.001), particularly in the temporalis (*p* < 0.001), masseter (*p* < 0.001), and TMJ regions (*p* < 0.001) (Table 3). A composite pain index, aggregating reports of temporalis, masseter, and TMJ pain, confirmed higher overall pain scores in sedentary individuals (*p* < 0.01).

Table 4 demonstrates correlation between the reported symptoms and the pain site, noise, deviation or deflection, and pain on palpation according to the level of physical activity and the presence of TMD.

Table 5 demonstrates correlation between the opening amplitude, lateral motion, and protrusion according to the level of physical activity and presence of TMD.

The Bonferroni correction was applied, adjusting the significance threshold. After correction, the associations between sedentary individuals (with and without TMD) and pain-free unassisted opening remained statistically significant (*p* ≤ 0.017).

A logistic regression analysis was performed to assess the association between physical activity status (sedentary vs. athlete) and reported pain (binary outcome). Results demonstrated that higher values of the diagnosis variable (indicative of more severe TMD or sedentary status) increased the likelihood of pain reporting (β = −0.1168, *p* < 0.001). The pseudo R^2^ of 0.1628 indicates that the model explains 16% of the variance in pain presence, which is reasonable for clinical and epidemiological studies. The Likelihood Ratio Test (LLR *p*-value = 2.03 × 10^−7^) confirmed the model’s statistical significance.

The logistic regression equation used in this study was log(p/(1 − p)) = 2.3667 − 0.1168 × Diagnostic, where p represents the probability of reporting pain. Diagnostic is a categorical variable representing sedentary vs. athlete status.

Although the pseudo R^2^ value (16%) indicates moderate explanatory power, the significant negative β-coefficient (−0.1168, *p* < 0.001) suggests that higher levels of physical activity are associated with a lower likelihood of reporting pain.

## 4. Discussion

This study provides compelling evidence of the link between lifestyle, particularly comparing the benefits and effects of regular training routines of CrossFit^®^ and the prevalence and severity of TMD symptoms and signs. It emphasizes the need for a broader perspective in TMD management, one that incorporates lifestyle modifications alongside traditional treatments for a more effective approach to patient care.

TMDs, being musculoskeletal disorders, affect a significant number of people and can lead to major health issues [28]. The prevalence of TMD in the Brazilian population was higher in females (37.0%) than in males (29.3%) [29]. In athletes, TMD is common, particularly in women, and has a multifactorial aspect [20]. The influence of practicing sports on the perception of signs and symptoms of TMD is directly related to the influence of psychosocial factors on athletes [21]. A deeper understanding of how to diagnose, treat, and monitor TMD symptoms can enhance patient care and improve outcomes. Focusing on the specific roles of these practitioners in TMD management—ranging from physical therapy interventions to dental treatments and sports-related injury management—can lead to more effective, multidisciplinary approaches and better patient satisfaction. Based on this, the hypothesis of this study was confirmed: the frequency and severity of TMD signs and symptoms were different between CrossFit^®^ athletes and sedentary individuals, in which sedentary individuals exhibited significantly higher prevalence rates of TMD compared to CrossFit^®^ athletes, with sedentary individuals being 2.3 times more likely to develop TMD.

In our study, the presence of joint noises and pain reports significantly differed between groups, with sedentary individuals with TMD experiencing higher frequencies of lateral motion and protrusion noises and significantly greater pain intensity scores compared to athletes. The chosen sport for analysis was CrossFit^®^, known for its diverse components in each workout, including aerobic capacity, muscle strength and endurance, speed, coordination, agility, and balance [30]. The varied and high-intensity nature of CrossFit^®^ may contribute to improved overall musculoskeletal function, postural stability, and neuromuscular control, which could play a role in preventing or alleviating TMD symptoms. By enhancing muscle strength, flexibility, and coordination, such training may help reduce muscular imbalances and excessive strain on the temporomandibular joint. These aspects of exercise physiology are relevant to understanding the potential benefits of physical activity in TMD management. However, the increase in injuries was associated with 3.5 and 3.2 times greater odds when there was alternation between training loads and previous injuries, respectively. The most injured areas in CrossFit^®^ athletes were the shoulder (26%), spine (24%), and knee (18%) [31]. Thus, further investigations are needed to understand the influence of exercise regularity and high intensity, as well as its effects at the muscular and articular levels.

The mechanisms underlying these observations might be attributed to the protective effects of regular physical activity, which has been associated with reduced systemic inflammation and enhanced neuromuscular control, contributing to lower pain perception and improved TMJ function in athletes [32]. Physical activity’s role in reversing maladaptive neural changes and reducing systemic inflammation provides a compelling explanation for the lower incidence of TMD symptoms among athletes [32]. On the other hand, contact sports may act as a risk factor for the development of TMD [33], as they can cause trauma to the face and temporomandibular joints. Boxers may experience impaired jaw function, muscle sensitization, local neural sensitization, and headaches, with a TMD frequency of 77.77% [34]. In handball athletes, the TMD frequency was 45.00% [34]. The prevalence of TMDs in rugby athletes was 53.3%, higher than in non-athletes (14.3%) [35]. The overall injury rate was 15.2/1000 athlete-exposures in rugby versus 4.9/1000 athlete-exposures in football; there are similarities between the modalities for the most common types of injuries (sprains and concussions), locations (lower extremity and head), and mechanisms (direct contact) [36]. However, individualized investigations are crucial to understanding the impact of TMJ and related structures concerning performed activities, given the variety of stimuli. The overall prevalence of TMD signs and symptoms in specific populations, including karate-do and mixed martial arts (MMA) athletes [25], and ballerinas [37], were also examined. Therefore, the analysis of TMD signs and symptoms should be individualized according to the sport practiced to better understand the etiology, preventive measures, characteristics related to dysfunction, and treatment.

In addition, high-performance athletes showed a higher TMD prevalence than non-athletes [33]. According to Prasad et al. [20], moderate- to vigorous-intensity exercise can be accompanied by a high number of contractions of the masseter muscles, which possibly leads to teeth clenching. Hilgenberg-Sydney et al. [37] found a 78% overall TMD prevalence in ballerinas, with higher rates for disc displacement with reduction (57%) and arthralgia (35%). Bonotto et al. [25] investigated the prevalence of temporomandibular disorders in high-performance martial arts fighters and compared it with the prevalence in recreational athletes and non-athletes. They observed that 54.2% of high-performance karate-do athletes (12.5% myofascial pain and 45.8% disk displacement), 17.6% of recreational martial arts practitioners (5.9% myofascial pain and 11.8% disk displacement), 65.1% of high-performance mixed martial arts athletes (30.8% myofascial pain and 38.5% disk displacement), and 14.3% of non-practitioners (7.1% myofascial pain and 7.1% disk displacement) were diagnosed with TDM. These findings contrast with our study, where myofascial pain was the most common diagnosis among both sedentary individuals and athletes. The previous literature indicates that, in an Italian population, the prevalence of different TMD diagnoses was higher in muscle disorders (38.2%) [24], which aligns with our findings.

However, at the other end of the spectrum of sporting activities, the literature is still scarce on the relationship between a sedentary lifestyle and TMD. Saccomanno et al. [38] developed a study to analyze the influence of the pandemic on various aspects of human health (obesity, TMJ disorders, and dental and respiratory problems, such as OSAS), using a questionnaire via phone call. They found that 48.1% of the sample reported suffering from orofacial/TMJ pain, with over half (51.8%) of these individuals noting pain worsening during the pandemic, likely due to reduced physical activity [38]. According to the authors, a sedentary lifestyle and small sporting activities resulting from social restrictions may explain the worsening of orofacial pain/TMD symptoms. Our findings corroborate this, as sedentary individuals had significantly higher odds of reporting pain and were more likely to experience muscle and joint pain in comparison to athletes. Therefore, sedentary individuals presented more frequent symptoms of TMD, in agreement with Friewald et al. [5], who indicate a higher prevalence of TMD symptoms in non-athletes compared to competitive female athletes, further reinforcing the association between physical inactivity and increased TMD-related pain and dysfunction.

In CrossFit^®^ athletes, no significant differences were found in mandibular deviation or deflection, though joint noises differed significantly, suggesting a higher prevalence of internal derangement features in sedentary individuals with TMD. Disc displacement prevalence in adults varies from 18% to 41% [10,38], and is clinically evident when it interferes with TMJ movement [39]. The significantly higher occurrence of opening, lateral motion, and protrusion noises in sedentary individuals with TMD may be indicative of TMJ dysfunctions that require further biomechanical assessment. Disc displacement etiologies are diverse, including TMJ anatomical factors, mechanical force trauma, generalized joint hypermobility, lubrication changes, and trauma [39]. However, there is no conclusive evidence favoring any single factor as dominant in the onset of displacement [39]. In most cases, disc displacement with reduction is stable, asymptomatic, and a lifelong joint condition [32,39], typically not requiring treatment as the structures adapt well.

Athletes exhibited the highest amplitudes in mouth-opening movements, with higher pain-free unassisted opening indices than sedentary individuals, regardless of TMD diagnosis. Considering the normal standard for pain-free unassisted mandibular opening as greater than 40 mm [40], athletes in this study showed average mouth openings within normal ranges (athletes without TMD = 41.50 ± 5.36 mm; athletes with TMD = 40.87 ± 7.49 mm), unlike the sedentary volunteers (sedentary without TMD = 32.38 ± 8.05 mm; sedentary with TMD = 34.82 ± 10.83 mm). Dinsdale et al. [41], in a systematic review and meta-analysis, reported that maximum active mouth opening was reduced by 4.65 mm in TMD groups compared to controls. Similar findings were noted in a group of ballerinas by Hilgenberg-Sydney et al. [37]. Although no significant difference was found in lateral mandibular movements between groups, sedentary individuals with TMD exhibited the lowest mean protrusive movement values, supporting previous reports that reduced protrusion is common in TMD individuals. Dinsdale et al. [41] also found no significant difference in laterotrusion between TMD and control groups, but a reduced protrusion in TMD individuals compared to controls.

In pain assessment, higher frequencies of pain reporting and palpation response were observed in sedentary individuals compared to athletes, regardless of TMD status. Chronic pain is recognized as a lifestyle-related disease because it is associated with a sedentary and inactive lifestyle [42]. Pain, especially when persistent, can negatively interfere with an athlete’s performance, making it difficult to maintain their competitive level. Physical activity appears to have a protective effect against pain and in performing excursive movements in athletes. This protective mechanism is associated with the nervous system’s plasticity, reversing maladaptive changes [43], and includes the analgesic effect of physical activity, which increases serotonin release, activating the inhibitory descending pain pathway [44,45]. Physical activity also has an anti-inflammatory effect, reducing systemic inflammation [45], while a sedentary lifestyle can increase inflammatory cytokines and decrease anti-inflammatory neurotransmitters like serotonin [46]. Regular physical exercise is required to maintain these benefits [45,47]. Healthy adults can experience ‘exercise-induced analgesia’, reducing pain perception during and after exercise [45,48], particularly with high-intensity aerobic exercise [48]. Excessive training can induce trauma to skeletal muscle, connective tissue, and bone, which promotes the release of pro-inflammatory cytokines [49]. The effects of the excessive release of cytokines resulting from exercise, besides affecting performance, can also cause pathological conditions in various structures, such as skeletal muscles, the hypothalamus, the liver, and the heart, suggesting a multi-organ model [50]. The etiology of TMD is complex and remains poorly understood, but several biological and psychosocial risk factors have been identified [1]. Increased sedentary behavior can result in increased levels of stress, anxiety, and depression [51].

The limitations of this study include the psychosocial assessment provided by axis II of the RDC/TMD, which was not applied. Additionally, the RDC/TMD was not replaced by the Diagnostic Criteria for Temporomandibular Disorders (DC/TMD) during the experimental phase. Future research should explore diverse populations and examine the effects of various physical activity types and intensities on TMD. It should also identify the most beneficial exercises and their mechanisms of action on TMD symptoms. Including factors like height, weight, and skeletal pattern could clarify the relationship between physical activity levels and mouth opening range. Additionally, studies on the long-term impact of lifestyle changes and the role of exercise duration and intensity in distinguishing athletes from sedentary individuals would be valuable. Understanding prevalence requires consideration of population characteristics, necessitating observational studies with longitudinal follow-up, which is a limitation of this research.

For over two decades, the RDC/TMD has generated substantial international scientific research, providing a strong evidence base for reliable and valid revisions [52]. A core principle underlying this diagnostic approach emphasizes using epidemiological data to discern the distribution of signs and symptoms by sex and age and to identify population norms for better disease definition [52]. Since this research predated the validation of the DC/TMD in Portuguese and primarily aimed to identify differences in TMD symptoms and signs between CrossFit^®^ athletes and sedentary individuals, we believe that the use of the RDC/TMD did not interfere with data collection validity. Compared to the RDC/TMD, the DC/TMD has advanced into an evidence-based system with greater validity for clinical use [52].

The implications of these findings are significant for clinical practice. They underscore the need for a holistic approach in managing TMD that includes not only clinical interventions but also lifestyle modifications, particularly emphasizing the importance of physical activity. Healthcare professionals should consider advising patients with TMD to engage in appropriate physical exercises as part of their treatment plan. In the present research, the results are associated with CrossFit^®^ practice and sedentary individuals.

## 5. Conclusions

This study showed that pain in the masseter, temporalis, posterior digastric, and medial pterygoid muscles—whether isolated or combined with other symptoms—was the main complaint in individuals with TMD, both athletes and sedentary. However, sedentary individuals reported greater pain frequency and intensity, as well as more pronounced limitations in mouth movement. These results indicated that a sedentary lifestyle may have worsened TMD symptoms, particularly pain sensitivity and functional impairment in the TMJ and masticatory muscles. In contrast, athletes with TMD were more likely to present joint noises, suggesting different TMD patterns depending on activity level.

## Figures and Tables

**Figure 1 ijerph-22-00785-f001:**
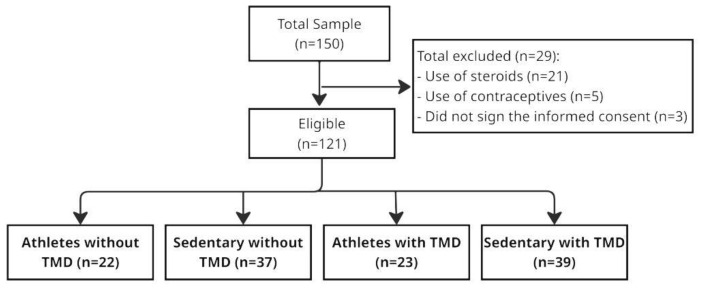
STROBE flowchart of sample selection and distribution into groups.

**Table 1 ijerph-22-00785-t001:** Distribution and comparison of the frequencies of the presence of deviation or deflection and noise according to the level of physical activity and the presence of TMD.

Variables	Athletes Without TMD	Sedentary Without TMD	Athletes with TMD	Sedentary with TMD	*p*-Value
**Presence of deviation or deflection**					0.13
Absent	2 (9.1%)	5 (13.5%)	0 (0.0%)	1 (2.6%)	
Present	20 (90.9%)	32 (86.5%)	23 (100.0%)	38 (97.4%)	
**Opening noises**					**<0.001**
Absent	18 (81.8%)	26 (70.3%)	5 (21.7%)	20 (51.3%)	
Present	4 (18.2%)	11 (29.7%)	18 (78.3%)	19 (48.7%)	
**Closing noises**					0.069
Absent	19 (86.4%)	32 (86.5%)	16 (69.6%)	25 (64.1%)	
Present	3 (13.6%)	5 (13.5%)	7 (30.4%)	14 (35.9%)	
**Reciprocal click eliminated during protrusive opening**					**0.027**
Absent	22 (100.0%)	37 (100.0%)	21 (91.3%)	33 (84.6%)	
Present	0 (0.0%)	0 (0.0%)	2 (8.7%)	6 (15.4%)	
**Lateral motion noises**					**<0.001**
Absent	17 (77.3%)	35 (94.6%)	8 (34.8%)	22 (56.4%)	
Present	5 (22.7%)	2 (5.4%)	15 (65.2%)	17 (43.6%)	
**Protrusion noises**					**0.002**
Absent	21 (95.5%)	32 (86.5%)	12 (52.2%)	29 (74.4%)	
Present	1 (4.5%)	5 (13.5%)	11 (47.8%)	10 (25.6%)	

Legend: The *p*-value from the chi-square test is *p* ≤ 0.05. Analyses with statistical significance are highlighted in bold.

**Table 2 ijerph-22-00785-t002:** Description and comparison of means, standard deviations, medians, and interquartile ranges (IQRs) of the variables of opening, lateral motion, and protrusion according to the level of physical activity and presence of TMD.

Variables	Athletes Without TMD (n = 22)		Sedentary Without TMD (n = 37)		Athletes with TMD (n = 23)		Sedentary with TMD (n = 39)		*p*-Value
	Mean (SD)	Median (IQR)	Mean (SD)	Median (IQR)	Mean (SD)	Median (IQR)	Mean (SD)	Median (IQR)	
Unassisted opening without pain (mm)	41.50 (5.36) ^a^	41.50 (39.00–45.00)	32.38 (8.05) ^b^	32.00 (27.50–35.00)	40.87 (7.49) ^a^	39.00 (35.00–48.00)	34.82 (10.83) ^b^	35.00 (27.00–44.00)	**<0.001**
Max. unassisted opening (mm)	52.91 (6.19)	52.00 (49.75–55.00)	49.24 (6.63)	49.00 (44.50–55.00)	50.70 (8.76)	50.00 (45.00–59.00)	47.28 (11.42)	50.00 (42.00–55.00)	0.061
Max. assisted opening (mm)	55.32 (7.00)	54.50 (51.00–59.25)	52.35 (6.04)	53.00 (48.00–57.50)	53.39 (8.81)	53.00 (47.00–62.00)	50.46 (10.85)	54.00 (45.00–57.00)	0.12
Right lateral motion (mm)	8.68 (1.84)	9.00 (7.00–10.00)	8.54 (2.88)	8.00 (6.50–10.50)	9.48 (3.19)	10.00 (8.00–12.00)	8.59 (3.48)	8.00 (6.00–10.00)	0.90
Left lateral motion (mm)	7.77 (1.54)	8.00 (7.00–9.00)	8.03 (2.70)	8.00 (6.00–10.00)	7.70 (2.53)	8.00 (6.00–9.00)	8.49 (3.56)	8.00 (6.00–10.00)	0.84
Protrusion (mm)	5.77 (2.02) ^a^	6.00 (4.75–6.00)	4. 68 (2.44) ^ab^	3.00 (5.00–6.00)	5. 48 (3.13) ^ab^	5.00 (4.00–7.00)	4.28 (2.96) ^b^	4.00 (2.00–5.00)	**0.039**

Legend: The *p*-value considered from the one-way ANOVA test is *p* ≤ 0.05. Statistically significant results appear in bold. Lowercase letters are used to indicate statistically significant differences between the groups.

**Table 3 ijerph-22-00785-t003:** Distribution and comparison of pain reporting frequencies, indication of the pain site, and response to palpation according to the level of physical activity and the presence of TMD.

Variables	Athletes Without TMD	Sedentary Without TMD	Athletes with TMD	Sedentary with TMD	*p*-Value
**Report of pain**					**<0.001**
Absent	22 (100.0%)	27 (73.0%)	13 (56.5%)	6 (15.4%)	
Present	0 (0.0%)	10 (27.0%)	10 (43.5%)	33 (84.6%)	
**Indication of pain in muscle and/or TMJ by the participant**					**<0.001**
Absent	22 (100.0%)	37 (100.0%)	13 (56.5%)	3 (7.7%)	
Present	0 (0.0%)	0 (0.0%)	10 (43.5%)	36 (92.3%)	
**Temporal pain**					**<0.001**
Absent	22 (100.0%)	36 (97.3%)	16 (69.6%)	13 (33.3%)	
Present	0 (0.0%)	1 (2.7%)	7 (30.4%)	26 (66.7%)	
**Masseter pain**					**<0.001**
Absent	21 (95.5%)	35 (94.6%)	16 (69.6%)	4 (10.3%)	
Present	1 (4.5%)	2 (5.4%)	7 (30.4%)	35 (89.7%)	
**Posterior digastric pain**					**<0.001**
Absent	20 (90.9%)	36 (97.3%)	16 (69.6%)	14 (35.9%)	
Present	2 (9.1%)	1 (2.7%)	7 (30.4%)	25 (64.1%)	
**Medial pterygoid pain**					**<0.001**
Absent	19 (86.4%)	35 (94.6%)	17 (73.9%)	12 (30.8%)	
Present	3 (13.6%)	2 (5.4%)	6 (26.1%)	27 (69.2%)	
**TMJ pain**					**<0.001**
Absent	21 (95.5%)	36 (97.3%)	18 (78.3%)	12 (30.8%)	
Present	1 (4.5%)	1 (2.7%)	5 (21.7%)	27 (69.2%)	
**Temporal tendon pain**					**<0.001**
Absent	19 (86.4%)	34 (91.9%)	14 (60.9%)	13 (33.3%)	
Present	3 (13.6%)	3 (8.1%)	9 (39.1%)	26 (66.7%)	

Legend: *p*-value based on chi-square test. Statistical significance level considered at *p* ≤ 0.05. Statistically significant results appear in bold.

**Table 4 ijerph-22-00785-t004:** Association of the report and indication of the pain site, noise, deviation or deflection, and pain on palpation according to the level of physical activity and the presence of TMD.

Variables	OR (95%CI)	*p*-Value
**Report of pain**		
Sedentary vs. athletes	9.53 (3.16–28.78)	**<0.001**
With TMD vs. without TMD	19.24 (6.63–55.85)	**<0.001**
**Indication of pain in muscle and/or TMJ by the participant**		
Sedentary vs. athletes	15.60 (3.70–65.69)	**<0.001**
With TMD vs. without TMD	-	-
**Presence of deviation or deflection**		
Sedentary vs. athletes	0.53 (0.10–2.81)	0.45
With TMD vs. without TMD	8.29 (0.98–69.77)	0.052
**Opening noises**		
Sedentary vs. athletes	0.64 (0.29–1.43)	0.28
With TMD vs. without TMD	4.41 (2.02–9.64)	**<0.001**
**Closing noises**		
Sedentary vs. athletes	1.17 (0.48–2.89)	0.73
With TMD vs. without TMD	3.27 (1.31–8.14)	**0.011**
**Reciprocal click eliminated during protrusive opening**		
Sedentary vs. athletes	1.91 (0.35–10.36)	0.45
With TMD vs. without TMD	-	-
**Lateral motion noises**		
Sedentary vs. athletes	0.33 (0.13–0.82)	**0.016**
With TMD vs. without TMD	8.98 (3.38–23.85)	**<0.001**
**Protrusion noises**		
Sedentary vs. athletes	0.65 (0.26–1.62)	0.35
With TMD vs. without TMD	4.58 (1.69–12.43)	**0.003**
**Temporal pain**		
Sedentary vs. athletes	4.74 (1.58–14.20)	**0.005**
With TMD vs. without TMD	82.53 (10.42–653.92)	**<0.001**
**Masseter pain**		
Sedentary vs. athletes	13.51 (3.88–47.07)	**<0.001**
With TMD vs. without TMD	83.64 (19.11–366.15)	**<0.001**
**Posterior digastric pain**		
Sedentary vs. athletes	2.64 (0.97–7.18)	0.057
With TMD vs. without TMD	21.74 (6.02–78.52)	**<0.001**
**Medial pterygoid pain**		
Sedentary vs. athletes	3.19 (1.20–8.51)	**0.021**
With TMD vs. without TMD	13.85 (4.72–40.60)	**<0.001**
**TMJ pain**		
Sedentary vs. athletes	6.04 (1.97–18.55)	**0.002**
With TMD vs. without TMD	39.64 (8.47–185.43)	**<0.001**
**Temporal tendon pain**		
Sedentary vs. athletes	1.98 (0.79–4.99)	0.15
With TMD vs. without TMD	11.99 (4.43–32.47)	**<0.001**

Legend: OR: odds ratio. CI: confidence interval. Statistical significance level considered at *p* ≤ 0.05. Statistically significant results appear in bold.

**Table 5 ijerph-22-00785-t005:** Association of the variables of opening amplitude, lateral motion, and protrusion according to the level of physical activity and presence of TMD.

Variables	(95%CI) *	*p*-Value
**Unassisted opening without pain**		
Athletes without TMD	Ref.	
Sedentary without TMD	−0.12 (−0.17; −0.07)	**<0.001**
Athletes with TMD	−0.01 (−0.05; 0.03)	0.62
Sedentary with TMD	−0.10 (−0.16; −0.03)	**0.005**
**Max. unassisted opening**		
Athletes without TMD	Ref.	
Sedentary without TMD	−0.03 (−0.06; −0.003)	**0.033** ^§^
Athletes with TMD	−0.02 (−0.06; 0.02)	0.25
Sedentary with TMD	−0.06 (−0.12; −0.01)	**0.032** ^§^
**Max. assisted opening**		
Athletes without TMD	Ref.	
Sedentary without TMD	−0.02 (−0.05; 0.004)	0.097
Athletes with TMD	−0.02 (−0.06; 0.02)	0.35
Sedentary with TMD	−0.05 (−0.10; 0.001)	0.055
**Right lateral motion**		
Athletes without TMD	Ref.	
Sedentary without TMD	−0.02 (−0.09; 0.05)	0.55
Athletes with TMD	0.003 (−0.11; 0.11)	0.96
Sedentary with TMD	−0.02 (−0.09; 0.05)	0.53
**Left lateral motion**		
Athletes without TMD	Ref.	
Sedentary without TMD	−0.003 (−0.07; 0.07)	0.94
Athletes with TMD	−0.03 (−0.13; 0.07)	0.54
Sedentary with TMD	0.01 (−0.07; 0.09)	0.83
**Protrusion**		
Athletes without TMD	Ref.	
Sedentary without TMD	−0.10 (−0.19; −0.01)	**0.040 ^§^**
Athletes with TMD	−0.05 (−0.17; 0.06)	0.36
Sedentary with TMD	−0.17 (−0.31; −0.04)	**0.013**

Legend: * simple linear regression with dependent variables log-transformed. CI: confidence interval. Statistical significance level considered at *p* ≤ 0.05. Statistically significant results appear in bold. ^§^ Loss of significance after correction for multiple testing (*p* ≤ 0.017).

## Data Availability

No new data were created or analyzed in this study. Data sharing is not applicable to this article. Further inquiries can be directed to the corresponding author(s).

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
