# Peer review of "Comparison of Temporomandibular Disorder Signs and Symptoms in CrossFit® Athletes and Sedentary Individuals"

_ijerph, 2025, doi:10.3390/ijerph22050785_

Round 1
Reviewer 1 Report
Comments and Suggestions for Authors
The article presents an interesting and in-depth analysis of the prevalence and differences in signs and symptoms of Temporomandibular Disorder (TMD) between CrossFit® athletes and sedentary individuals. The following are some suggestions to improve the clarity, coherence, and overall understanding of the article:
Abstract
Background
Please specify why it is relevant to compare CrossFit® athletes to sedentary individuals in the context of TMD. The current statement is too general. Please explicitly mention the role of physical activity in pain modulation and joint function.
Methods
Please clarify the composition of the sample. Please specify the total number of participants and their distribution across the four groups. If possible, please include numerical details (e.g., "n = X per group") to improve methodological transparency.
Additional information on variables measured: The use of Research Diagnostic Criteria for TMD (RDC/TMD) is mentioned, but it would be helpful to specify the parameters assessed (e.g., muscle pain, range of motion, joint noises).
Results
Include key numeric values ​​Providing precise measurements and p-values ​​strengthens the results section. Instead of stating that athletes had greater jaw ranges of motion and lower pain levels, include precise data.
Distinguish groups more clearly: indicate whether athletes with TMD had fewer symptoms than sedentary individuals with TMD or whether differences were only observed between athletes and sedentary individuals overall.
Conclusion
The conclusion actually reiterates the findings. It highlights practical implications and suggests preventive or therapeutic strategies. It highlights the potential protective role of physical activity in the management of TMD and offers directions for future research.
Manuscript
In the Background section, mention how TMD can present with symptoms such as earache, tinnitus, chewing difficulties, and dizziness, as evidenced in the Messina studies. "Previous studies have associated TMD with symptoms such as ear pain, tinnitus, difficulty chewing, and dizziness, suggesting a complex interrelationship between these conditions." (https://pubmed.ncbi.nlm.nih.gov/32499885/)
In the study design section or the results section, it would be helpful to include a clearer description of the key characteristics of the groups (e.g., age, gender, level of education, etc.) for easier comparison.
Although the article presents the results, there is little in-depth discussion of the clinical implications of these findings (lines 278-284). It would be helpful to explore in more detail the practical aspects for healthcare professionals, such as physical therapists, dentists, or sports medicine physicians.
The Limitations and Future Directions section (lines 403-417) could be expanded. For example, greater attention to the duration and intensity of physical activity could help to better understand the differences between athletes and sedentary individuals.
Discuss the possible association between an elongated styloid process and TMD symptoms, as described by Messina. "An elongated styloid process has been correlated with TMD symptoms, suggesting a possible anatomical interrelation that warrants further investigation." (https://pubmed.ncbi.nlm.nih.gov/32499885/)
While the article describes the findings in detail, the conclusion (lines 409-417) could be strengthened by summarizing the main key points more clearly. For example, by emphasizing the importance of promoting physical activity as a preventive measure against TMD.
Author Response
Reviewer 1
Comment 1: The article presents an interesting and in-depth analysis of the prevalence and differences in signs and symptoms of Temporomandibular Disorder (TMD) between CrossFit® athletes and sedentary individuals. The following are some suggestions to improve the clarity, coherence, and overall understanding of the article:
Response 1: Thank you for your thoughtful feedback and suggestions to improve the clarity and coherence of the article. We truly appreciate your time and contributions.
- Abstract
Background
Comment 2: Please specify why it is relevant to compare CrossFit® athletes to sedentary individuals in the context of TMD. The current statement is too general. Please explicitly mention the role of physical activity in pain modulation and joint function.
Response 2: Thanks for pointing this out. We agree with this comment and the new information was added in the abstract. The information is below:
“Physical exercise is recommended for pain management and improving quality of life. Comparing CrossFit® athletes to sedentary individuals allows to examine whether regular high-intensity exercise impacts pain sensitivity and functional limitations associated with TMD.”.
Methods
Comment 3: Please clarify the composition of the sample. Please specify the total number of participants and their distribution across the four groups. If possible, please include numerical details (e.g., "n = X per group") to improve methodological transparency.
Response 3: We appreciate the comment. The information was added as requested. It is below:
“Participants (n=121) were divided into four groups: sedentary with TMD (n=39), sedentary without TMD (n=37), CrossFit® athletes with TMD (n=23), and CrossFit® athletes without TMD (n=22).”
Comment 4: Additional information on variables measured: The use of Research Diagnostic Criteria for TMD (RDC/TMD) is mentioned, but it would be helpful to specify the parameters assessed (e.g., muscle pain, range of motion, joint noises).
Response 4: Thank you for your comment. We will provide additional details on the specific parameters assessed using the Research Diagnostic Criteria for TMD (RDC/TMD).
“…including mandibular movement patterns, range of motion, joint sounds, muscle pain, and jaw dysfunctions”.
Results
Comment 5: Include key numeric values Providing precise measurements and p-values strengthens the results section. Instead of stating that athletes had greater jaw ranges of motion and lower pain levels, include precise data.
Distinguish groups more clearly: indicate whether athletes with TMD had fewer symptoms than sedentary individuals with TMD or whether differences were only observed between athletes and sedentary individuals overall.
Response 5: Thanks for the comments. We have added the suggestions as requested. The rewritten sentences are below:
“All athlete groups exhibited greater mandibular movement amplitudes (unassisted opening without pain, p<.001, and protrusion, p=.039) and less pain (p<.001) than sedentary individuals.”
“Pain reports and palpation-induced pain in muscles and joints were significantly associated with a sedentary lifestyle and TMD (p<.001).”
Conclusion
Comment 6: The conclusion actually reiterates the findings. It highlights practical implications and suggests preventive or therapeutic strategies. It highlights the potential protective role of physical activity in the management of TMD and offers directions for future research.
Response 6: Thanks for your comment. We appreciate your feedback, and we are glad that you found the conclusion effectively highlights the practical implications, as well as the potential protective role of physical activity in managing TMD.
- Manuscript
Comment 7: In the Background section, mention how TMD can present with symptoms such as earache, tinnitus, chewing difficulties, and dizziness, as evidenced in the Messina studies. "Previous studies have associated TMD with symptoms such as ear pain, tinnitus, difficulty chewing, and dizziness, suggesting a complex interrelationship between these conditions." (https://pubmed.ncbi.nlm.nih.gov/32499885/)
Response 7: Thank you for your valuable suggestion. We have accepted your recommendation and incorporated the information into the Background section of the manuscript. The reference indicated has also been added.
Comment 8: In the study design section or the results section, it would be helpful to include a clearer description of the key characteristics of the groups (e.g., age, gender, level of education, etc.) for easier comparison.
Response 8: Thanks for this consideration. The first paragraph was rewritten as requested.
“The final sample comprised 121 participants, of whom 58.67% were female (Athletes without TMD: 45, Sedentary without TMD: 55.88, Athletes with TMD: 48, and Sedentary with TMD: 77.77). The average age was ± 29 years. Most sedentary individuals were aged 18 to 27 (without TMD: 68.75% and with TMD: 72.23%). For athletes, both groups showed 40% in the 28 to 37 age range. The majority of participants were single (72.25%) and had incomplete higher education. Most sedentary individuals had incomplete higher education (without TMD: 64.7% and with TMD: 78.57%). Among athletes, 70% of those without TMD had completed higher education, while 60% of those with TMD had completed high school. The study population was divided into sedentary individuals (n=76) and CrossFit® athletes (n=45).”
Comment 9: Although the article presents the results, there is little in-depth discussion of the clinical implications of these findings (lines 278-284). It would be helpful to explore in more detail the practical aspects for healthcare professionals, such as physical therapists, dentists, or sports medicine physicians.
Response 9: Thanks for the relevant comment. We agreed and wrote the following sentences:
“A deeper understanding of how to diagnose, treat, and monitor TMD symptoms can enhance patient care and improve outcomes. Focusing on the specific roles of these practitioners in TMD management—ranging from physical therapy interventions to dental treatments and sports-related injury management—can lead to more effective, multidisciplinary approaches and better patient satisfaction.”.
Comment 10: The Limitations and Future Directions section (lines 403-417) could be expanded. For example, greater attention to the duration and intensity of physical activity could help to better understand the differences between athletes and sedentary individuals.
Response 10: Thanks for the relevant comment. We agreed and wrote the following sentences:
"Greater attention to the duration and intensity of physical activity could help to better understand the differences between athletes and sedentary individuals."
Comment 11: Discuss the possible association between an elongated styloid process and TMD symptoms, as described by Messina. "An elongated styloid process has been correlated with TMD symptoms, suggesting a possible anatomical interrelation that warrants further investigation." (https://pubmed.ncbi.nlm.nih.gov/32499885/)
Response 11: Thank you for your insightful suggestion. The focus of our study did not specifically consider the styloid process, which is why this aspect was not included in the discussion. We acknowledge that the elongated styloid process, as highlighted by Messina, has been associated with TMD symptoms, suggesting a potential anatomical link that merits further exploration. While this correlation was not within the scope of our current research, it is an important area for future studies to address, as it could provide valuable insights into the multifactorial nature of TMD.
Comment 12: While the article describes the findings in detail, the conclusion (lines 409-417) could be strengthened by summarizing the main key points more clearly. For example, by emphasizing the importance of promoting physical activity as a preventive measure against TMD.
Response 12: Thank you for your thoughtful suggestion. The authors have discussed your recommendation and, after reviewing the article by Weiler et al. (2013) [Prevalence of signs and symptoms of temporomandibular dysfunction in female adolescent athletes and non-athletes], we decided not to add the proposed emphasis on promoting physical activity as a preventive measure against TMD in the conclusion. The referenced study concluded that physical activity did not prevent the onset of TMD. Additionally, our work did not focus on the incidence of TMD, but rather on other aspects of the condition. We appreciate your input and will certainly consider it in future research directions.
We sincerely thank you for the detailed and constructive review. I inform you that all the suggested changes have been made to the article.
Reviewer 2 Report
Comments and Suggestions for Authors
Dear Authors,
Thank you for your submission. Your study provides valuable insights into the relationship between physical activity and temporomandibular disorders (TMD), particularly in CrossFit athletes versus sedentary individuals. Below are some suggestions to enhance clarity, presentation, and methodological transparency.
Methods & Results
- Provide a more detailed explanation of the measures taken by the research team to minimize bias.
- Clearly specify the multiple comparison correction methods applied (e.g., Bonferroni correction).
- It would be beneficial to provide height and weight information for the sedentary group and the athlete group. Since mouth opening range can vary depending on height, if there is a significant difference in height between the sedentary and athlete groups, it could affect the explanatory power of the results.
Discussion
- Consider factors such as diet, stress levels, and previous injury history, as they may also influence TMD symptoms.
- Provide a more detailed discussion on the mechanisms by which exercise may help prevent or alleviate TMD symptoms, as this would enhance the study’s clinical applicability.
The paper is well written overall, but some sentences are overly complex and long, making them difficult to read. If these are organized into more concise sentences, readers will be able to understand the content more easily. If you check the grammar and sentence structure one more time and polish them naturally, the quality of the paper will be further improved.
Author Response
Reviewer 2
Dear Authors,
Thank you for your submission. Your study provides valuable insights into the relationship between physical activity and temporomandibular disorders (TMD), particularly in CrossFit athletes versus sedentary individuals. Below are some suggestions to enhance clarity, presentation, and methodological transparency.
Response: We appreciate the considerations that you have made. We considered them to improve our manuscript.
Methods & Results
- Comment 1: Provide a more detailed explanation of the measures taken by the research team to minimize bias.
Response 1: Thanks for this consideration. We consider such important. A new paragraph was created to explain in details the measures to minimize bias as requested.
“To minimize bias in this clinical study, observers were standardized in their evaluation procedures and data collection methods. They were blinded to the data distribution during tabulation, ensuring impartiality in the analysis. Additionally, a researcher from a non-dental field was selected to further reduce potential bias, as they had no prior knowledge of the meaning of the collected data, the study outcomes, or the hypothesis under investigation. Furthermore, the results were analyzed separately for different subgroups within the total sample, enhancing the reliability and robustness of the findings.”.
- Comment 2: Clearly specify the multiple comparison correction methods applied (e.g., Bonferroni correction).
Response 2: Thank you for your valuable feedback. We have now explicitly stated the multiple comparison correction method used in our analysis. This clarification has been incorporated into the manuscript in the Methods section.
“Post-hoc pairwise comparisons were performed using Bonferroni correction to adjust for multiple comparisons [27], and control the Type I error rate.”
- Comment 3: It would be beneficial to provide height and weight information for the sedentary group and the athlete group. Since mouth opening range can vary depending on height, if there is a significant difference in height between the sedentary and athlete groups, it could affect the explanatory power of the results.
Response 3: Thank you for the suggestion. We would like to inform you that the data were collected according to the analysis instrument used, the RDC/TMD questionnaire, Axis I. However, since there was no specific indication to collect height and weight information, this focus was not included in the approach regarding the association with mouth opening range. We acknowledge that these variables may influence the mouth opening range, and based on your suggestion, we plan to include, in future studies, the correlation between sedentary and athletic status and mouth opening range, considering factors such as weight, height, and other related aspects, such as skeletal pattern.
“Future studies should consider including height, weight, and other relevant factors, such as skeletal pattern, to better understand the correlation between sedentary and athletic status and mouth opening range.”
Discussion
Comment 4: Consider factors such as diet, stress levels, and previous injury history, as they may also influence TMD symptoms.
Response 4: Thank you for the suggestion. We would like to inform you that the data collected were obtained from Axis I of the RDC/TMD questionnaire, which refers to the clinical examination. Since there was no indication to collect information about diet and trauma history, these factors were not addressed in the study sampling. As for psychosocial factors, these are collected using Axis II of the mentioned questionnaire. Based on your suggestion, we have added the following statement to the article:
"Additionally, studies exploring the long-term effects of lifestyle changes on TMD progression would be valuable."
- Comment 5: Provide a more detailed discussion on the mechanisms by which exercise may help prevent or alleviate TMD symptoms, as this would enhance the study’s clinical applicability.
Response 5: Thank you for your suggestion. The second paragraph of the Discussion section was reformulated to attend the request.
“The varied and high-intensity nature of CrossFit may contribute to improved overall musculoskeletal function, postural stability, and neuromuscular control, which could play a role in preventing or alleviating TMD symptoms. By enhancing muscle strength, flexibility, and coordination, such training may help reduce muscular imbalances and excessive strain on the temporomandibular joint. These aspects of exercise physiology are relevant to understanding the potential benefits of physical activity in TMD management.”.
We sincerely thank you for the detailed and constructive review. I inform you that all the suggested changes have been made to the article.
Reviewer 3 Report
Comments and Suggestions for Authors
This manuscript studies the relationship between the practice of Crossfit and the apparition of TMD compared with sedentary population. The English seems correct but I am not native, so I suggest another reviewer for this.
Some issues need to be solved before processing further:
-Conclusion in the abstract is not clear, it does not answer the objectives: “Myofascial pain was the most prevalent condition” cannot be a conclusion as this is not a prevalence study.
-All cites must be before de dot, inside the sentence which are referred. Apply to the whole text.
-Line 39: please define that “heterogneous group of conditions”.
-Line 41-43: author do not provide enough information about the factors involved.
-Line 69: the relationship between Crossfit and why these subjects could suffer from TMD disorders is not clear at all.
-Line 74. Please cite those “scarce” studies.
-Line 90: please add date of approval.
-Line 104: according to these inclusion criteria, all participants have TMD disorders, so no healthy group were recruited.
-Conclusion have part of discussion, while it must sucintly answer the objective of the study.
Author Response
Reviewer 3
This manuscript studies the relationship between the practice of Crossfit and the apparition of TMD compared with sedentary population. The English seems correct but I am not native, so I suggest another reviewer for this.
Response: Thank you for your feedback. We appreciate your suggestion, and a thorough English language review has already been conducted to ensure clarity and accuracy. However, we remain open to further revisions if necessary.
Some issues need to be solved before processing further:
Comment 1: -Conclusion in the abstract is not clear, it does not answer the objectives: “Myofascial pain was the most prevalent condition” cannot be a conclusion as this is not a prevalence study.
Response 1:Thanks for this appointment. The conclusion was rewritten as requested.
“Myofascial pain is the most common TMD condition in athletes and sedentary individuals. However, sedentary individuals with TMD experience higher frequency and intensity of pain, as well as greater limitations in mouth movement. Athletes showed higher frequency of joint noises.”
Comment 2: -All cites must be before de dot, inside the sentence which are referred. Apply to the whole text.
Response 2: Thanks for this appointment. All the corrections were made.
Comment 3: -Line 39: please define that “heterogneous group of conditions”.
Response 3: Thank you for your suggestion. By "heterogeneous group of conditions," we refer to the fact that Temporomandibular Disorders (TMD) encompass a variety of distinct conditions that can affect different structures, including the jaw, masticatory muscles, and temporomandibular joint (TMJ). These conditions can vary in their symptoms, causes, and clinical presentations, making TMD a complex and multifactorial condition.
Comment 4: -Line 41-43: author do not provide enough information about the factors involved.
Response 4: Thank you for your suggestion. The factors were added as requested.
"These disorders can significantly impact individuals' quality of life and have a multifactorial etiology, manifesting signs and symptoms due to both pathophysiological (systemic, local, and genetic factors) and psychosocial (stress, anxiety, limited coping skills, depression, etc.) factors".
Comment 5: -Line 69: the relationship between Crossfit and why these subjects could suffer from TMD disorders is not clear at all.
Response 5: Thanks for your comment. We added more information about it.
“CrossFit® athletes, while performing repetitive movements that lead to exhaustion, may acquire harmful oral behaviors that can be considered etiological factors for TMD (bruxism, jaw muscle fatigue, improper posture, and increased stress).”
Comment 6: -Line 74. Please cite those “scarce” studies.
Response 6: Thanks for this consideration. The references [5,19,21] were added in the referred sentence.
Comment 7: -Line 90: please add date of approval.
Response 7: Thanks for this consideration. We inform you that the date of approval was added in the first paragraph of the Materials and Methods section.
Comment 8: -Line 104: according to these inclusion criteria, all participants have TMD disorders, so no healthy group were recruited.
Response 8: Thank you for your comment. The study design considered sedentary individuals, both with and without TMD, as the comparison control group. This approach was implemented to facilitate a comparison with CrossFit athletes, both with and without TMD, allowing for a more accurate analysis of the differences between these groups. Additionally, the other inclusion and exclusion criteria ensured that all participants in the study were in good health, further validating the conditions of the groups being compared.
Comment 9: -Conclusion have part of discussion, while it must sucintly answer the objective of the study.
Response 9: The reviewer’s comment has been acknowledged. The conclusion has been reorganized to succinctly address the study's objective. The sections containing discussion-related content have been moved to the appropriate discussion section.
We sincerely thank you for the detailed and constructive review. I inform you that all the suggested changes have been made to the article.
Round 2
Reviewer 3 Report
Comments and Suggestions for Authors
I want to thank the authors for their efforts implementing my suggestions. However, some points must have been misunderstood, so I must emphasize:
-Comment 1: -Conclusion in the abstract is not clear, it does not answer the objectives: “Myofascial pain was the most prevalent condition” cannot be a conclusion as this is not a prevalence study. What I mean is that this manuscript is not about prevalence of TMD disorders nor (according to the objectives) it tries to determine which disorder is more prevalent. This manuscript is a comparison between two populations, NOT a prevalence study, so determining as conclusion that “Myofascial pain was the most prevalent condition” when authors have not considered other conditions, is not acceptable.
-
Comment 3: -Line 39: please define that “heterogneous group of conditions”.
Response 3: Thank you for your suggestion. By "heterogeneous group of conditions," we refer to the fact that Temporomandibular Disorders (TMD) encompass a variety of distinct conditions that can affect different structures, including the jaw, masticatory muscles, and temporomandibular joint (TMJ). These conditions can vary in their symptoms, causes, and clinical presentations, making TMD a complex and multifactorial condition.
ANSWER: I perfectly understood what you mean, I just pointed the need to define that in the manuscript, but I have seen that you added no information to clarify this in the text, so I must insist.
Comment 8: -Line 104: according to these inclusion criteria, all participants have TMD disorders, so no healthy group were recruited.
Response 8: Thank you for your comment. The study design considered sedentary individuals, both with and without TMD, as the comparison control group. This approach was implemented to facilitate a comparison with CrossFit athletes, both with and without TMD, allowing for a more accurate analysis of the differences between these groups. Additionally, the other inclusion and exclusion criteria ensured that all participants in the study were in good health, further validating the conditions of the groups being compared.
ANSWER: I must have mis-explained myself, so please accept my apologies. What I mean is that inclusion criteria are not clear and them must be re-written in a clearer way. You used to paragraphs to explain them (116-122 and 123-133) and in one you stated "
The inclusion criteria for all participants were as follows: diagnosis of muscular TMD,
which could be associated with joint TMD" while in the next paragraph you determined that "
The inclusion criteria were also considered sedentary volunteers as individuals who
did not do any physical activity in any sport, either amateur or professional for at least six
months." Inclusion criteria must be in a single paragraph and describes clearly enough to understand what was done about this, so please check this.
I must congratulate the authors for the rest of implementations and hope that this time I have clearly indicated the way the issues could be solved.
Author Response
Dear reviewer,
We sincerely thank you for your valuable comments and suggestions. We apologize for any misunderstanding or lack of clarity regarding some of the points raised in your previous review. We have carefully revised the manuscript and made the necessary corrections in accordance with your observations. We trust that the changes will address the issues appropriately.
-Comment 1: -Conclusion in the abstract is not clear, it does not answer the objectives: “Myofascial pain was the most prevalent condition” cannot be a conclusion as this is not a prevalence study. What I mean is that this manuscript is not about prevalence of TMD disorders nor (according to the objectives) it tries to determine which disorder is more prevalent. This manuscript is a comparison between two populations, NOT a prevalence study, so determining as conclusion that “Myofascial pain was the most prevalent condition” when authors have not considered other conditions, is not acceptable.
- Response: Thank you for pointing this out. We agree with this comment. The conclusion was rewritten as requested.
“Pain in the masseter, temporalis, posterior digastric, and medial pterygoid muscles was the most common symptom in sedentary individuals with TMD”
Comment 3: -Line 39: please define that “heterogneous group of conditions”.
Response 3: Thank you for your suggestion. By "heterogeneous group of conditions," we refer to the fact that Temporomandibular Disorders (TMD) encompass a variety of distinct conditions that can affect different structures, including the jaw, masticatory muscles, and temporomandibular joint (TMJ). These conditions can vary in their symptoms, causes, and clinical presentations, making TMD a complex and multifactorial condition.
ANSWER: I perfectly understood what you mean, I just pointed out the need to define that in the manuscript, but I have seen that you added no information to clarify this in the text, so I must insist.
- Response: We appreciate your comment. The sentence was rewritten as requested.
“Temporomandibular disorders (TMD) are a heterogeneous group of conditions, that is, it includes several clinical conditions (pain localized in muscles of mastication or preauricular area), each requiring distinct management approaches, which is aggravated with chewing or other mandibular activities or movements [1-3].”
Comment 8: -Line 104: according to these inclusion criteria, all participants have TMD disorders, so no healthy group were recruited.
Response 8: Thank you for your comment. The study design considered sedentary individuals, both with and without TMD, as the comparison control group. This approach was implemented to facilitate a comparison with CrossFit athletes, both with and without TMD, allowing for a more accurate analysis of the differences between these groups. Additionally, the other inclusion and exclusion criteria ensured that all participants in the study were in good health, further validating the conditions of the groups being compared.
ANSWER: I must have mis-explained myself, so please accept my apologies. What I mean is that inclusion criteria are not clear and them must be re-written in a clearer way. You used to paragraphs to explain them (116-122 and 123-133) and in one you stated "
The inclusion criteria for all participants were as follows: diagnosis of muscular TMD,
which could be associated with joint TMD" while in the next paragraph you determined that "
The inclusion criteria were also considered sedentary volunteers as individuals who
did not do any physical activity in any sport, either amateur or professional for at least six
months." Inclusion criteria must be in a single paragraph and describes clearly enough to understand what was done about this, so please check this.
- Response: Thank you for pointing this out. We agree with this comment. As requested, we reorganized the paragraph to be concise and complete.
“The inclusion criteria for all participants were: age between 18 and 48 years; absence of rheumatological conditions, neurological disorders, motor impairments, primary headaches, or orofacial inflammatory/infectious conditions [20]; no regular use of medication; no toothache, polyarthritis, or other rheumatic diseases; and provision of informed consent. For participants with temporomandibular disorders (TMD), only those diagnosed with muscular TMD—with or without associated joint TMD—were included. Sedentary individuals were defined as those who had not engaged in any amateur or professional sport or physical activity for at least six months. Sedentary behavior was characterized as low-energy expenditure while awake, in various settings such as work, school, home, community, or during transportation [22]. CrossFit® practitioners were required to have competed in regional and/or national competitions, trained at least four times per week for a minimum of six months, and attended an accredited facility under the supervision of a certified coach. Their training regimen involved intermittent and cyclical exercises of varying intensities, including weightlifting, plyometrics, calisthenics, running, and indoor rowing [12].”